# Generation of Endotoxin-Specific Monoclonal Antibodies by Phage and Yeast Display for Capturing Endotoxin

**DOI:** 10.3390/ijms25042297

**Published:** 2024-02-15

**Authors:** Alexandra C. Fux, Cristiane Casonato Melo, Laura Schlahsa, Nico B. Burzan, André Felsberger, Isabel Gessner, Jonathan A. Fauerbach, Jutta Horejs-Hoeck, Miriam Droste, Christiane Siewert

**Affiliations:** 1Department of Biosciences and Medical Biology, Paris Lodron University of Salzburg, 5020 Salzburg, Austria; 2Research and Development Department, Miltenyi Biotec B.V. & Co. KG, 51429 Bergisch Gladbach, Germany

**Keywords:** endotoxin, antibodies, phage display, yeast display, scFv-Fc, microbeads, LPS removal, flow cytometry, LAL, cell sorting

## Abstract

Endotoxin, a synonym for lipopolysaccharide (LPS), is anchored in the outer membranes of Gram-negative bacteria. Even minute amounts of LPS entering the circulatory system can have a lethal immunoactivating effect. Since LPS is omnipresent in the environment, it poses a great risk of contaminating any surface or solution, including research products and pharmaceuticals. Therefore, monitoring LPS contamination and taking preventive or decontamination measures to ensure human safety is of the utmost importance. Nevertheless, molecules used for endotoxin detection or inhibition often suffer from interferences, low specificity, and low affinity. For this reason, the selection of new binders that are biocompatible, easy to produce, and that can be used for biopharmaceutical applications, such as endotoxin removal, is of high interest. Powerful techniques for selecting LPS-binding molecules in vitro are display technologies. In this study, we established and compared the selection and production of LPS-specific, monoclonal, human single-chain variable fragments (scFvs) through two display methods: yeast and phage display. After selection, scFvs were fused to a human constant fragment crystallizable (Fc). To evaluate the applicability of the constructs, they were conjugated to polystyrene microbeads. Here, we focused on comparing the functionalized beads and their LPS removal capacity to a polyclonal anti-lipid A bead. Summarized, five different scFvs were selected through phage and yeast display, with binding properties comparable to a commercial polyclonal antibody. Two of the conjugated scFv-Fcs outperformed the polyclonal antibody in terms of the removal of LPS in aqueous solution, resulting in 265 times less residual LPS in solution, demonstrating the potential of display methods to generate LPS-specific binding molecules.

## 1. Introduction

Sepsis affects more than 30 million people globally and is responsible for over 5 million deaths each year [1]. The main cause of sepsis is exposure to pyrogenic substances, such as the Gram-negative bacterial cell wall component called endotoxin, which is also known as lipopolysaccharide (LPS). Even low levels of LPS can trigger strong immune reactions in the human circulatory system, resulting in proinflammatory cytokine secretion (e.g., IL-6, IL-1β, and TNFα) [2,3]. Regarding its structure, LPS comprises three major regions: the hydrophilic O-Antigen, consisting of a variable number of up to 50 saccharide monomers and facing towards the extracellular space, the hydrophobic lipid A, which functions as an anchoring unit to the cell wall, and the core, which connects these two regions [3]. Approximately 75% of the outer bacterial cell wall consists of LPS [4]. Considering that LPS is released during bacterial division, as well as growth and cell death, the endotoxin load in the environment is high, making it an omnipresent hazardous substance and a common source for contamination, affecting both solutions and surfaces. Therefore, the control and containment of endotoxin levels are inevitable when it comes to the medical safety of pharmaceuticals. Regulatory agencies such as the U.S. Food and Drug Administration (FDA) and the European Medicines Agency (EMA) published strict guidelines regarding the permissible levels of endotoxins in injectable formulations and medical devices. These levels are measured in endotoxin units (EUs) and correspond to 0.1 ng of LPS from *Escherichia coli* (*E. coli*), according to the FDA. Moreover, endotoxin limits depend on various parameters such as the contact point (e.g., blood, cardiovascular system, or spinal fluid) and use (e.g., injectable medicine, permanent implant, and subcutaneous implant) [5]. These limits are likely to be lowered in the future [6]. Thus, the selection of LPS-binding molecules that allow for the detection and/or capture of LPS is of high interest.

To date, various LPS-binding molecules are known [3,7]. However, many of them face drawbacks such as non-specificity [7] or are too expensive for economical assay development (e.g., commercially available LPS-binding protein) [8]. Moreover, LPS tends to non-specifically adhere to a multitude of surfaces [9,10], proteins, and other molecules, obstructing a consistent orientation. Due to its amphiphilic nature, it readily forms supramolecular structures [11,12], shielding individual molecules from potential binders. Hence, the generation of LPS-specific binders poses a major obstacle on the way to efficient LPS removal assays.

Methods to remove LPS from solutions involve the use of LPS-removal platforms such as fibers [13] or resins [14] covered with various LPS-binding units. However, LPS removal in a non-specific manner tends to unintentionally remove proteins of interest as well [15]. With these challenges in mind, targeting LPS has been described in multiple studies [7] and has been in focus over the last few years, expanding LPS removal to novel platforms such as nanoparticles or microbeads [16,17,18].

As we recently demonstrated [18], efficient LPS removal from different complex solutions can be achieved by using bead-based removal assays in combination with a polyclonal anti-lipid A (aLA) antibody. In the work presented here, we aim to take this removal assay to the next level by using monoclonal antibodies in order to increase capture and removal capacities. Monoclonal antibodies, in general, bind their corresponding antigen more specifically and at single epitopes [19], potentially being more reliable and a superior option for specific LPS removal. In addition, monoclonal antibodies allow for high-scale production with greater purity due to high expression levels in vitro without the need for animal inoculation. Traditional methods for monoclonal antibody production, such as immunizing animals and screening antibodies using, e.g., hybridoma technology [19], might pose challenges due to the high toxicity of immunoreactive targets [20].

A highly effective method for in vitro binder screening is phage display, a molecular biology technique wherein peptides or antibodies are presented on the surface of bacteriophages, allowing the selection of specific binding molecules through successive rounds of enrichment and amplification on immobilized targets [21,22]. Moreover, human naïve libraries, derived from non-immunized donors, provide a basis for targeting toxic antigens [23]. In general, natural IgG antibodies are perfect candidates as binding molecules since they combine a stable backbone region consisting of the fragment crystallizable (Fc) and a highly specific antigen-binding region. This binding region is formed by a heavy (VH) and light (VL) variable region. Artificially fusing VH and VL with a linker creates a single-chain variable fragment (scFv) [24]. Connecting scFv with the constant Fc region forms a scFv-Fc construct, a reduced, synthetic antibody (approximately 110 kDa vs. 150 kDa for IgG). Advantages of the scFv-Fc construct in contrast to natural IgG are a simpler cloning process as well as higher production yields [25]. In addition, the Fc region’s binding capability towards protein A allows for a straightforward purification.

Recent studies present the production of LPS-binding molecules in the form of Fab (fragment antigen binding) [26] or peptides [27,28] via phage display. However, since phage display involves expression in bacteria, especially *E. coli*, targeting LPS can pose difficulties due to unwanted interactions with intrinsic LPS. Hence, instead of using *E. coli* as a phage multiplicator, switching to a yeast-based platform circumvents this hurdle.

In contrast to phage display, cells for yeast display are genetically engineered *Saccharomyces cerevisiae* (*S. cerevisiae*) [29], which are devoid of LPS. In this study, a yeast display library was firstly incubated with LPS, our target molecule. Secondly, scFv-expressing yeast with strong antigen-binding capacity are sorted via fluorescence-activated cell sorting (FACS). A noteworthy difference in this selection process when working with engineered yeast is the use of antigens in solution rather than immobilizing LPS on a surface. This approach might reduce the risk of steric hindrance of the binders binding to LPS in contrast to utilizing a flat well plate during phage display. Additionally, FACS sorting enables the selection of high-affinity binders by placing a sort gate in an adjustable and desired position.

Ultimately, our aim is to generate and implement binders that can be used in bead-based LPS-related applications, such as removal or detection assays. In this study, we describe the generation and comparison of LPS-binding scFv-Fc via phage and yeast display. To our knowledge, this work is the first to employ yeast display in order to create scFv-Fc for LPS-specific binding. Furthermore, we explore the differences between our monoclonal antibodies and a commercially available polyclonal aLA antibody. Finally, we demonstrate the applicability of our generated binders in a bead-based LPS removal assay.

## 2. Results and Discussion

The aim of this study was to generate monoclonal antibodies (scFv-Fc) against *E. coli* LPS serotype O111:B4 by using two different display methods. The generated antibodies are intended to serve as a basis for bead-based LPS removal and detection assays.

### 2.1. Screening and Evaluation of LPS Monoclonal Antibodies via Phage Display

As a starting point for the generation of monoclonal antibodies, two human, naïve scFv-M13 filamentous bacteriophage sublibraries were used in this study, each composed of different types of the VL domains (κ or λ). The library was developed in-house by Miltenyi Biotec, composed of naïve B-cell isolations from more than 120 healthy donors, resulting in an antibody diversity of 5 × 10^10^. Figure 1a shows a schematic overview of each work step performed in the phage display selection strategy. The scFv-presenting phage library was incubated on immobilized biotinylated LPS from *E. coli* O111:B4 on streptavidin-coated plates. According to the manufacturer, biotinylation of LPS was located at the hydrophilic O-Antigen; thus, the combination of streptavidin coating and LPS-biotin favored an oriented approach, enhancing the accessibility of the phage particles to the core, phosphate groups, and acyl chains. A similar concept of emphasizing LPS orientation was previously described [30]. After three panning cycles, eluted LPS-specific phage particles were enriched in *E. coli* TG1, as shown in Figure 1b. The number of λ-VL colonies containing infective phage particles increased from 1.2 × 10^5^ after the first round to 8.2 × 10^6^ after the third round, and for κ-VL colonies containing infective phage particles, from 6.3 × 10^4^ to 5.6 × 10^5^.

To screen a great number of clones, six 96-well plates were inoculated with randomly picked *E. coli* TG1 colonies (576 clones) to produce soluble scFv. The supernatant was used to screen the binding capability of biotinylated LPS in comparison to non-specific binding to streptavidin/bovine serum albumin (BSA), as shown in Figure 1c. Out of 576 clones, twelve clones were found to have streptavidin background absorbance signals (A_450_–A_630_) below 0.06 and LPS signals (A_450_–A_630_) above 0.12. Sequencing results revealed that some of these twelve clones recurred, leading to only three unique amino acid sequences (clones P1, P2, and P3). The distribution of the twelve clones corresponded here to 50% for clone P1, 8.3% for clone P2, and 41.7% for clone P3. Signal differences between identical clones can be explained by a variance between the amount of *E. coli* in the overnight culture and the resulting difference in the produced quantity of soluble scFv. Therefore, the signal strength of the clones at this point is not a quantitative measure but rather a qualitative indication of LPS binding. P1 and P2 possessed a λ-VL, and P3 was derived from a κ-VL. P3 was later excluded since it bound streptavidin at higher antibody concentrations, as shown in Appendix A. Experiments were continued with clones P1 and P2. The scFv clones were cloned into an expression vector including the Fc region and produced in Expi293F human embryonic kidney (HEK) cells. The produced scFv-Fcs were analyzed using the SDS-PAGE method, as shown in Figure 1d. The primary bands were at a height of around 130–150 kDa. Faint bands were neglected.

To evaluate the LPS-binding performance towards LPS from *E. coli* serotype O111:B4, the two selected scFv-Fcs (P1 and P2) were compared to the commercially available polyclonal aLA IgG as a positive LPS-binding antibody control in ELISA. Additionally, a randomly selected scFv-Fc clone was used as a negative control. As shown in Figure 1e, background (streptavidin/blocking) absorbance values (A_450_–A_630_) were subtracted from biotinylated LPS signals (A_450_–A_630_). Hereby, the highest LPS binding signal at low scFv-Fc concentrations can be attributed to P1. In general, binding capabilities towards LPS showed stronger signals for P1 than for the polyclonal aLA. EC_50_ values are shown in Appendix A.

### 2.2. Screening and Evaluation of LPS Monoclonal Antibodies via Yeast Display

To generate an antibody gene library in *S. cerevisiae*, a preselection via phage display on LPS was performed. By harvesting phage plasmids instead of the phage supernatant from phage display panning round one, contact between potential binders and *E. coli* debris was avoided. The obtained genes were cloned into a yeast display vector. Using this newly created library, we screened for further LPS binders via yeast display. This approach allows for a more direct selection of positive binders due to the possibility of sorting cells by gating high-affinity binders.

#### 2.2.1. Combining Phage and Yeast Display

The scFv selection process against LPS from *E. coli* O111:B4 using yeast display was carried out as shown in Figure 2a. As a primary source, the same phage library was used, and the procedure was performed as described in Section 2.1 with two exceptions. The two sublibraries (λ and κ) were mixed before incubating on biotinylated LPS, and two different blocking buffers were used. As BSA shows interactions with LPS [31], alternative blocking reagents were used to decrease masking effects on possible binding sites for the scFv clones. Therefore, casein (CN) and sheep serum (SS) were used as an alternative since animal sera were found to be superior in regard to blocking unspecific binding sites [32]. As shown in Figure 2b, more than 4 × 10^5^ infective phage particles were eluted after the panning round for both approaches.

Phage panning is performed in *E. coli* and therefore has the disadvantage that scFvs presented by phages might bind to the cell wall or cell debris (including free LPS), which is released overnight due to the growth, division, or death of *E. coli*. To avoid contact of LPS with promising binders before the next selection step, plasmids of infected *E. coli* were therefore isolated and cloned (as shown in Figure 2c) in the target vector pYSDM2. To generate homologous overhangs for recombinant cloning in yeast, the scFvs were amplified out of pYSDM2. Linearized pYSDM2 and scFv-insert ligations of both strategies (CN and SS, as shown in Figure 2c) were mixed and electroporated into *S. cerevisiae* strain EBY100, reaching a total CFU of 1.13 × 10^7^.

#### 2.2.2. Selection for LPS-Specific scFv Clones via Cell Sorting

The scFv-expressing *S. cerevisiae* were prepared in several steps before sorting for the best LPS binders, as shown in Figure 3a. Starting with a total of 1 × 10^8^ cells, cells were first incubated with magnetic streptavidin beads for 1h to remove any unwanted streptavidin binders by magnetic separation. Remaining cells (flow-through) were stained with anti-HA-PE since all scFv-expressing yeast possessed a HA tag, as shown in Figure 3a. After staining with anti-HA-PE antibody-fluorochrome, cells were incubated with anti-PE MicroBeads and subsequently positively selected for the scFv clones. To further increase the purity of the scFv-expressing cells, magnetically labeled cells were enriched by a subsequent round of positive selection. Hereby, starting from 74% scFv-expressing cells, these were enriched to 98% on the first magnetic column and to over 99% in the final second purification step, as shown in Appendix A. After this step, the enriched scFv-expressing cells were incubated with biotinylated LPS for 1 h at RT and stained with Streptavidin-APC for identification of LPS-binding cells.

Labeled *S. cerevisiae* cells were sorted in MACSQuant Tyto for the brightest APC signal (LPS binding), as shown in Figure 3b (red gates). The gated population was retained and cultured on agar plates until new clones were grown over 3–4 days. Clones were then pooled, and the procedure shown in Figure 3a was repeated. In the next two rounds, we increased our overall purity by sorting in two consecutive flow sorts. As shown in Appendix A, in round two, we enriched our starting population from 10% to over 70% of LPS-binding yeast cells. The gated population was expanded for the next round by plating them on agar plates. Subsequently, emergent clones were pooled and prepared for the next sorting round. The third round was carried out according to the same procedure, this time enriching only the very bright cells (9%) from the input fraction to over 77%. The enriched population was grown on agar plates for another 3–4 days, and 288 single clones were picked. Then, the obtained antibody genes were sequenced and narrowed down by choosing only recurrent sequences, leading to 30 unique clones, which were then cloned into a scFv-Fc format and screened individually for their LPS binding signals, as shown in Appendix A. Hereby, three binders were selected (Y1, Y2, and Y3), all originating from the λ-derived sublibrary. For additional characterization of LPS binding in a soluble form (LPS-biotin), scFv-expressing *S. cerevisiae* clones Y1, Y2, and Y3 were individually screened by flow cytometry, as shown in Figure 3c, displaying high median fluorescence intensity (MFI) signals for Y3. The gating strategy for the evaluation of staining intensity is shown in Appendix A. Selected binders were produced on a larger scale, purified, and applied to an SDS-PAGE gel, as shown in Figure 3d. The main protein bands at around 140 kDa corresponded to the scFv-Fc and appeared at a similar height as expected due to their similar molecular weight. Additionally, the observed weak bands at around 60 kDa (suspected single chains of scFv-Fc) and around 100 kDa (scFv-Fcs lacking glycosylation) were considered negligible.

### 2.3. Conjugation of scFv-Fc and Anti-Lipid A on Polystyrene Beads

Conjugation of scFv-Fc and polyclonal aLA antibody on aminodextran-functionalized 3.2-μm PS beads was carried out according to the same procedure as described in a previous study [18]. The remaining protein in the supernatant of the conjugation procedure was assessed using bicinchoninic acid (BCA) Reducing Agent Compatible. Hence, the amount of conjugated antibody was deduced to be 5.8 μg on 1 × 10^8^ beads for the polyclonal aLA beads. This was comparable to the 5.1 ± 0.6 μg per 1 × 10^8^ of 3-μm beads previously reported [18]. The amount of antibody for the different scFv-Fc beads was in the range of 7–13 μg per 1 × 10^8^ beads. The quality of the conjugation of the scFv-Fc was additionally assessed by flow cytometry using anti-human-Fc-PE and anti-goat-PE for detection of the scFv-Fc and anti-LA antibodies, respectively. More than 99.9% of all antibody (Ab)-conjugated beads were positive in the PE channel compared to non-conjugated beads as a gating control (Appendix A), thereby confirming the conjugation of the antibodies to the beads.

### 2.4. LPS-Removal Capacity of scFv-Fc Beads Versus Anti-Lipid A Beads

The LPS-removal capacity of the commercially available polyclonal aLA antibody that was conjugated to PS beads was evaluated by Casonato Melo et. al. [18]. Hereby, the authors incubated 10,000 EU/mL of standard LPS from *E. coli* O111:B4 in twelve consecutive cycles with aLA-PS beads. After 60,000 EU/mL, they observed LPS saturation on their aLA-PS beads, and therefore a stronger decline in LPS removal occurred for the following LPS incubation cycles. To directly compare the LPS-removal capacity of the polyclonal aLA bead with different scFv-Fc-beads, we chose to incubate 60,000 EU/mL LPS from *E. coli* O111:B4 with 5 mg/mL antibody-beads (Ab beads). A limulus amoebocyte lysate (LAL) assay was used to measure the remaining LPS concentration in the treated solution. The amount of endotoxin was quantified based on a freshly prepared calibration curve and measured together with the samples (exemplary data in Appendix A). As shown in Table 1, all beads showed an LPS-removal capability of at least 98.22%, confirming that this approach was effective in the conditions tested. Figure 4a shows the remaining LPS that was left after the incubation of 60,000 EU/mL with each bead type. According to this test, yeast-derived scFv-Fc clones left on average eight times less remaining LPS than the phage-derived antibodies (see also Appendix A). A possible explanation could be that the yeast clones were screened using solubilized rather than immobilized LPS as compared to phage display. Since phage display screening was performed on a plastic surface, yeast display-derived binders might be advantageous when binding LPS in solution. Since polyclonal aLA antibodies were generated by the immunization of goats, the same advantages might apply. Comparing the LPS-removal capability of the aLA-PS beads against Y1-PS and Y3-PS beads, the yeast display-derived Ab beads left 265 and 53 times less remaining endotoxin under the same condition, respectively. Overall, all Ab beads showed a remarkable removal capacity, whereas Y1-PS and Y3-PS beads showed the highest LPS removal under the tested conditions.

### 2.5. LPS Binding via Flow Cytometry

To assess the LPS binding ability of the conjugated PS beads according to the same conditions as explained for the LPS removal experiment in Section 2.4, different concentrations of biotinylated LPS were incubated with antibody-conjugated beads in water. Blocking the beads with 10% FBS for 1 h reduced background signals before staining the beads with Streptavidin-APC. MFI of APC was calculated for each LPS concentration ranging from 0 to 200 kEU/mL (see Figure 4b). The gating strategy for the determination of the MFI of APC is shown in Appendix A. The scFv-Fc beads Y1-PS and Y3-PS captured the highest amounts of LPS, as visualized by the strong APC signals. Table 2 summarizes the EC_50_ of all six antibody-coupled beads from lowest to highest (Y1 > Y3 > P1 > aLA > Y2 > P2). The results correlated well with the LPS-removal results from Section 2.4, where Y1 and Y3 showed the highest LPS-removal as well. Interestingly, the aLA-PS bead captured less LPS on its surface in comparison to the other Ab beads. Unconjugated PS beads were used as a negative control. Given the hydrophobic nature of the unconjugated PS bead surface, non-specific interactions with the hydrophobic LPS molecule can occur. Therefore, they can capture LPS even in the absence of specific binders to a certain degree, stressing the utility of using PS beads as a control for non-specific binding.

## 3. Materials and Methods

### 3.1. Materials

#### 3.1.1. Lipopolysaccharides

Biotinylated LPS from *E. coli* O111:B4 (#tlrl-lpsbiot) was purchased from InvivoGen (San Diego, CA, USA), and LPS from *E. coli* O111:B4 (#L2630-25M) was acquired from Sigma-Aldrich (St. Louis, MO, USA).

#### 3.1.2. Enzymes

NcoI-HF (#R3193), NotI-HF (#R3189), rSAP #M0371L, and T4 DNA Ligase (#M0202) were acquired from New England BioLabs (Ipswich, MA, USA). Anti-human IgG peroxidase (#A0170-1ML) was obtained from Sigma-Aldrich. Anti-goat-HRP (#A16148) and Phusion High-Fidelity DNA Polymerase (#F530L) were purchased from Thermo Fisher Scientific (Waltham, MA, USA).

#### 3.1.3. Cell Lines

NEB stable competent *E. coli* (#C3040) was purchased from New England BioLabs. *S. cerevisiae* EBY100 (#MYA-4941) was acquired from ATCC (Manassas, VA, USA). Expi293F (#A14527) was purchased from Thermo Fisher Scientific (Waltham, MA, USA).

#### 3.1.4. Plasmids

For the transformation of *S. cerevisiae* EBY100, the vector pYSDM2 used was adapted from pYSDM1 in [33]. Hereby, the order of the HA-tag was modified, placed next to the c-myc, and therefore a free amino terminus was created. The expression vector pFUSE-hIgG1-Fc (derived from pFUSE, InvivoGen (San Diego, CA, USA)) was used for the expression of scFv-Fc in Expi293F.

#### 3.1.5. Primers

Phage vector Fwd: CATGAAATACCTATTGCCTACGGPhage vector Rev: TGATGATGGTGATGATGGGAYeast pYSDM2 Primer Fwd: TCATGCAGTTACTTCGCTGTYeast pYSDM2 Primer Rev: CCTGCAGCAAGTCCTCTTC

#### 3.1.6. Antibodies

Anti-HA-PE (#130-120-717) and Streptavidin-APC (#130-106-791) were manufactured by Miltenyi Biotec (Bergisch Gladbach, Germany). Polyclonal goat anti-lipid A (aLA) antibody (#PA1-73178) was acquired from Thermo Fisher Scientific (Waltham, MA, USA). The secondary antibodies anti-human phycoerythrin (anti-human PE, #109-116-098) and anti-goat phycoerythrin antibody (anti-goat PE, #705-115-147) were acquired from Jackson ImmunoResearch (West Grove, PA, USA).

#### 3.1.7. Kits

NucleoSpin Gel and PCR Clean-up Kit (#740609) and NucleoSpin Plasmid (#740588) were obtained from Macherey-Nagel (Düren, Germany). Zymoprep-96 Yeast Plasmid Miniprep (#D2007) and ZymoPURE II Plasmid Midiprep Kit (#D4200) were purchased from Zymo Research (Irvine, CA, USA). Pierce Chromogenic Endotoxin Quant Kit (#A39553), Pierce BCA Protein Assay Kit—Reducing Agent Compatible (#23250), and ExpiFectamine 293 Transfection Kit (#A14525) were acquired from Thermo Fisher Scientific (Waltham, MA, USA). The Qiagen Plasmid Plus 96 Kit was purchased from Qiagen (Venlo, The Netherlands).

#### 3.1.8. Chemicals and Reagents

Carboxy polystyrene beads of 3.20 µm (PC05003, Lot #16082) were acquired from Bangs Laboratories, Inc. (Fishers, IN, USA). A single-chain fragment variable (scFv) phage library, sheep serum, and anti-myc-HRP were generated at Miltenyi Biotec (in-house, Bergisch Gladbach, Germany). LS Columns (#130-042-401), MACSQuant Tyto Running Buffer (#130-107-207), and MACSQuant Tyto Cartridges HS (#130-121-549) were acquired from Miltenyi Biotec (Bergisch Gladbach, Germany). Casein from bovine milk (#C5679), dextrose (#G7021), D-sorbitol (#S1876), Select Agar (#A5054), Trizma Base (#93350), Yeast Nitrogen Base (#Y0626), and trypsin (#T4424) were acquired from Sigma-Aldrich (St. Louis, MO, USA). Endotoxin-free water (#W50) was purchased from Lonza (Basel, Switzerland). HiTrap Protein A HP (#17040201), and PD-10 desalting columns (#17085101) were purchased from Cytiva (Marlborough, MA, USA). Ninety-six PhyTip Columns (#PTB 92-05-07) were acquired from PhyNexus (Uppsala, Sweden). For the SDS-PAGE gels, precast gels (4–20%, #5671094) and 4 × Laemmli Sample Buffer (#1610747) were purchased from Bio-Rad (Hercules, CA, USA). Der Blaue Jonas Protein Gel Stain (#GRP1) was purchased from German Research Products (Haag, Germany). TRIS-Glycine-SDS 10× running buffer (#TG50001) was purchased from anamed Elektrophorese (Groß-Bieberau, Germany). PageRuler Prestained Protein Ladder (#26616), GeneRuler 1 kb Plus (LT-02241), and Nunc MaxiSorb plates (#442404) were acquired from Thermo Fisher Scientific (Waltham, MA, USA). Corning FiltrEX 96-well plates (#3504) were purchased from Corning (Corning, NY, USA). EXIMUS Maximus FBS (#BS-2020-500) was acquired from Catus Biotech (Tutzing, Germany). Bovine serum albumin (BSA, #P6154) and streptavidin (#ROCKS000-01) were ordered from VWR (Radnor, PA, USA). IPTG (#2316.4), casein hydrolysate (#A157.3), sodium chloride (#3957.2), sodium dihydrogen phosphate dihydrate (#T879.2), and Tween20 (#9127.1) were purchased from Carl Roth (Karlsruhe, Germany). Agarose Standard for Electrophoresis (#9920.0500) was acquired from Th. Geyer (Lohmar, Germany). Veggie Peptone (#71280) and Veggie Yeast Extract (#71279) were purchased from Merck (Darmstadt, Germany). Di-sodium hydrogen phosphate anhydrous (#A2943) was purchased from PanReac AppliChem (Darmstadt, Germany). TMB (#S-100-TMB) was acquired from Seramun Diagnostica (Heidesee, Germany).

#### 3.1.9. Devices and Software

Flow cytometry was performed using a MACSQuant X Flow Cytometer and a MACSQuant 16 Analyzer. The obtained data were analyzed with FlowLogic version 7.3. Cell sorting experiments were performed on the MACSQuant Tyto Cell Sorter. A T100 thermal cycler from Bio-Rad (Hercules, CA, USA) was used for PCR steps. Äkta purifier 10 from GE Healthcare (Solingen, Germany) was used to purify the scFv-Fcs. Colonies were picked using the QPix Microbial Colony Picker. A Synergy H1 microplate reader was used to measure LAL results, and an Epoch microplate spectrophotometer from BioTek (Winooski, VT, USA) was used to measure ELISA with software Gen5 version 3.04. A Tecan HydroFlex 8-channel washer was used for the ELISA washing steps. ELISA Phage Display screening was analyzed using Vortex version 22.1.119916-2. Graphs were created using Graph Pad Prism 8.2.1. For the illustrations, Microsoft PowerPoint version 2308 and BioRender.com were used (accessed on 13 February 2024).

### 3.2. Methods

#### 3.2.1. Panning for LPS-Specific Binders via Phage Display

There were three rounds of panning in total. MaxiSorb plates were coated with 10 µg/mL streptavidin for 1 h and blocked with 2% BSA and PBS-T (PBS pH 7.4 + 0.05% Tween20) overnight. After washing three rounds with PBS-T, 10 µg/mL biotinylated LPS from *E. coli* strain O111:B4 was applied to the antigen wells; the counter wells remained without LPS. The human naïve single-chain fragment variable (scFv) phage library was first incubated on the counter wells (streptavidin/BSA/PBS-T) for 1 h at RT to diminish the amount of streptavidin or albumin binders. Hereby, sublibraries derived from λ and κ were processed on different plates. Subsequently, phages were transferred to the LPS-coated antigen wells. Additionally, 5 µg of streptavidin (as a competitor) were added, and phages were incubated for 2 h at RT. Wells were washed 10/20/30 times with PBS-T in rounds 1/2/3, respectively, and phages were eluted each time with 50 µL of 10 μg/mL trypsin, including incubation for 30 min at 37 °C. Subsequently, 1 mL of *E. coli* TG1 in 2×YP medium (1.6% peptone, 1% yeast extract, and 0.5% NaCl) were infected at an OD_600_ of 0.4–0.5 at 37 °C, rotating at 500 rpm. To measure phage titers, after 30 min of incubation, 10 µL were withdrawn and plated on 2×YP agar (2×YP medium, 100 mM glucose, and 100 µg/mL ampicillin, 15 g/mL agar). Another 30 min later, 1 M glucose and 1 mg/mL ampicillin stock solution were added in a volumetric ratio of 1:10 and incubated for 30 min at 37 °C and 500 rpm. Then, 3 × 10^10^ CFU M13K07 helper phages were added and incubated for 1 h at 37 °C at 500 rpm. As a last step, the cells were centrifuged at 4000 rpm for 10 min, and the supernatant was replaced with medium (2×YP medium supplemented with 50 µg/mL kanamycin and 100 µg/mL ampicillin). They were then incubated overnight at 30 °C, rotating at 500 rpm. On the next day, the cells were centrifuged (at 4000 rpm for 10 min), and the supernatant was harvested and used for the next panning round as a substitute for the phage library.

#### 3.2.2. ELISA Screening of LPS-Specific Binders

TG1 *E. coli* (OD_600_ of 0.4–0.5) were infected with phages from the third round and incubated for 30 min at 37 °C. Then, cells were plated on 2×YP agar dishes at 37 °C overnight. Single colonies were picked using a QPix Microbial Colony Picker into a 96-well format and incubated at 37 °C for 16 h at 300 rpm and 70% humidity. Subsequently, 10 µL of each well was transferred to a corresponding scFv production well plate containing 2×YP medium supplemented with 100 µg/mL ampicillin and 50 µM IPTG. Plates were incubated at 30 °C for 16 h at 300 rpm and 70% humidity.

ELISA screening was carried out in a 384-well plate, coated with 1 µg/mL streptavidin for 1 h at RT, and then blocked with PBS-T/BSA for 16 h. After washing the wells with PBS-T, they were incubated with biotinylated LPS for 1 h at RT or left as a negative control. Next, each well was incubated with 30 µL from the scFv production plate containing anti-myc-HRP for 1 h at RT. After washing the wells three times with PBS-T, 30 µL of TMB was added, and the plates were incubated for 10 min at RT. Then, the reaction was stopped using 30 µL of H_2_SO_4_. Plates were measured with the Epoch Microplate Spectrophotometer at A_450_–A_630_, and the data were analyzed using the software Vortex. The positive hits were sequenced by LGC genomics. Unique clones that showed intact scFv genes were selected for cloning. Plasmids from promising clones were isolated using the NucleoSpin Plasmid kit according to the manufacturer’s protocol.

#### 3.2.3. Cloning into a Fc-Expression Plasmid and Purification

Plasmid DNA from chosen clones was used as a PCR template to amplify scFvs out of the vector. The PCR constructs were purified using NucleoSpin Gel and the PCR Clean-up Kit, and afterwards, they were cleaved using NcoI and NotI and purified a second time. For the restriction digestion of the expression plasmid, rSAP was added for dephosphorylation of restriction sites to avoid re-ligation. Ligation was performed with the pFuseFc2.0-hIgG1 in an insert-to-vector ratio of 3:1 for 14–16 h at 16 °C. Plasmids were transformed into NEB *E. coli* in accordance with the manufacturer’s instructions and incubated overnight. To check for successful cloning, constructs were sequenced by LGC genomics. Expression vectors were then isolated and purified using the Qiagen Plasmid Plus 96 Kit. Antibodies were produced using HEK Expi293F cells in a volume of 0.8 mL (96-well format) according to the Gibco Expi 293 Expression System manual. Constructs were purified from the supernatant using ProPlus PhyTip Columns and Biomek NXp.

#### 3.2.4. Generation of a scFv-Expressing Yeast Library

The following method was inspired by publications [34,35] to create a combination of phage and yeast display. After one round of panning as described in the phage display Section 3.2.1, plasmids were isolated from the *E. coli* pellet on the following day instead of using the supernatant as described above. Constructs were then enzymatically cleaved with NcoI-HF and NotI-HF in rCutSmart buffer. For the restriction digestion of yeast plasmid pYSDM2, rSAP was added for dephosphorylation of restriction sites in the yeast plasmid to avoid re-ligation. Ligation was performed with an insert-to-vector ratio of 3:1 for 14–16 h at 16 °C. These ligated plasmids were used as a PCR template to generate scFv inserts with homologous overhangs for the yeast vector pYSDM2 for recombination cloning in yeast. The PCR protocol was performed as described in Table 3. For this, primers were used, which bound approximately 50 base pairs next to the scFv in the yeast vector. PCR products were purified using NucleoSpin Gel and PCR Clean-up. An agarose gel was run to check the constructs. GeneRuler 1 kb Plus was used as a standard. Linearized vector DNA and insert DNA were mixed in a ratio of 1:4 (4 µg:12 µg) with electrocompetent *S. cerevisiae*, which were generated with LiAc/DTT according to [36]. The yeast/DNA mixtures were electroporated at 2.5 kV, according to [36]. The transformed yeast cells were plated out on big agar dishes and incubated for 60 h at 30 °C. The cells were scraped down from the agar plates, collected in SD-CAA media and used as a yeast library for the selection process.

#### 3.2.5. Sorting for LPS-Specific Clones via the MACS Quant Tyto Cell Sorter

The medium and agar were prepared as follows:SD-CAA medium pH 4.5 sterile filtered (2% *w*/*v* dextrose, 0.6% *w*/*v* yeast nitrogen base, 0.5% casein hydrolysate, 0.74% *w*/*v* citric acid, and 1% sodium citrate).SD-CAA agar pH 6.0 (18.2% *w*/*v* sorbitol, 1.5% select agar, 0.54% *w*/*v* disodium phosphate, and 0.86% *w*/*v* monosodium phosphate—in 800 mL autoclaved) and added at 55 °C to sterile filtered 200 mL (0.67% *w*/*v* yeast nitrogen base, 0.5% *w*/*v*, and 2% *w*/*v* dextrose).SG-CAA medium pH 6.0 sterile filtered (1.8% *w*/*v* galactose, 0.2% *w*/*v* dextrose, 0.67% *w*/*v* yeast nitrogen base, 0.5% casein hydrolysate, 0.54% *w*/*v* disodium phosphate, and 0.86% *w*/*v* monosodium phosphate).

Yeast libraries (1 × 10^8^ cells) were incubated overnight at 30 °C and 250 rpm in SD-CAA medium. Cells were then passaged to an OD_600_ of 0.3–0.4 in fresh SD-CAA medium. At OD_600_ of 2–5, cells were centrifuged at 2500× *g* for 5 min and resuspended in a concentration of 1 × 10^7^ cells/mL in SG-CAA medium and incubated for at least 20 h at 20 °C and 250 rpm [34]. Then, a total of 1 × 10^8^ cells (1 mL of OD_600_ of 10) were centrifuged and washed in PBS-BSA. Cells were incubated for 1 h with streptavidin beads while rotating at RT. Subsequently, cells were magnetically separated in LS columns as described in the manufacturing manual (Miltenyi Biotec, Bergisch Gladbach, Germany). Continuing with the flow-through, cells were stained with aHA-PE (1:50) for 10 min in the dark on ice, washed, and incubated with anti-PE beads for magnetic cell separation in LS columns. The magnetic separation was performed twice to increase the purity of scFv-expressing yeast. Continuing with the positive-selected scFv-expressing yeast, cells were incubated with 170 nM biotinylated LPS for 1 h while rotating at RT. During the following steps, the reagents and samples were kept on ice. Cells were centrifuged at 4000× *g* for 5 min and washed with cold PBS-BSA. Then, the cells were stained with streptavidin-APC (1:50) for 10 min in the dark on ice. Subsequently, cells were washed three times and resuspended in MACSQuant Tyto running buffer, transferred to the cartridge input chamber, and sorted for the highest APC signal using a MACSQuant Tyto Cell Sorter. The yeast collected in the positive chamber was plated on SD-CAA agar plates and incubated for 2–4 days at 30 °C until clones appeared. Picked *S. cerevisiae* single clones were incubated overnight in SD-CAA medium. Plasmids were isolated using Zymoprep-96 Yeast Plasmid Miniprep according to the manufacturer’s instructions. They were then amplified for sequencing, performed by LGC genomics, and 30 recurring clones were chosen. The cloning procedure was carried out as shown in Section 3.2.3.

#### 3.2.6. Titration of Biotinylated LPS on Yeast Cells

Single yeast clones (Y1, Y2, and Y3) and wild-type *S. cerevisiae* as a control were inoculated overnight at 30 °C in SD-CAA and YPD medium (2% *w*/*v* glucose, 1% *w*/*v* yeast extract, and 2% *w*/*v* veggie peptone), respectively. On the next day, yeast clones were transferred to a fresh passage with an initial OD_600_ of 0.2. At an OD_600_ of 2–5, 1 × 10^8^ cells were centrifuged at 2500× *g* for 5 min, resuspended in SG-CAA to an OD_600_ of 0.5–1, and incubated for 16–20 h at 20 °C. Subsequently, the cells were washed with PBS by centrifuging at 11,000× *g* for 1 min and plated on a 96-well plate with 1 × 10^6^ cells/well. The cells were incubated with varying concentrations of biotinylated LPS (0; 100; 1000; 10,000; 50,000; 100,000; 200,000; and 300,000 EU/mL) for 1 h. Then, the cells were washed twice with PBS/BSA by centrifuging at 4000× g for 5 min and stained in a volume of 100 µL with anti-HA-PE (1:50) and streptavidin-APC (1:50) in PBS/BSA in the dark for 10 min. The cells were then washed twice with PBS/BSA and resuspended in 200 µL of PBS/BSA for measurement in flow cytometry (MACSQuant Analyzer 16). The data were analyzed using FlowLogic version 7.3. To determine the MFI, a gating strategy was used to remove debris and doublets from the analysis. Then, only the scFv-expressing cells (PE-positive) were gated and analyzed for LPS binding (APC-positive). The MFI of the APC-positive population was then determined.

#### 3.2.7. Antibody Production and Purification on a Larger Scale

After the selection of the five candidates, chosen antibodies were produced on a larger scale (200 mL volume) in HEK Expi293F cells following the manufacturers’ instructions in the Gibco Expi293 Expression System manual. For the purification step, scFv-Fc were extracted from the supernatant using HiTrap Protein A HP columns and Äkta purifier 10. To neutralize, 1 M TRIS at pH 9 was employed, and elution of the columns was carried out using 0.1 M sodium citrate at pH 3. PD-10 desalting columns were used to exchange elution buffer into PBS. The produced scFv-Fcs were mixed with 4 × Laemmli buffer in a ratio of 1:4 and denatured at 95 °C for 10 min. Samples were then loaded onto a precast SDS-PAGE gel submerged in TRIS-Glycine-SDS running buffer. The PageRuler Prestained Protein Ladder was used as a protein standard. The gel was run at 220 V and 60 mA for 50 min and then stained for 10 min using Der Blaue Jonas Protein Gel Stain, according to the manufacturer’s protocol.

#### 3.2.8. ELISA Measurements of scFv-Fc

Ninety-six-well plates were coated with streptavidin for 1 h and subsequently blocked overnight at 4–8 °C with PBS-T-BSA (0.05% Tween, 1% BSA). PBS-T (0.05%) was used to wash the wells after every incubation step, and all following incubation steps were carried out at RT if not stated otherwise. After washing the plates, the wells were incubated with 1 µg/mL biotinylated LPS from *E. coli* O111:B4 for 1 h at RT. Then, antibodies in different concentrations (0–500 nM) were incubated for 1 h in PBS. As controls, a non-specific scFv-Fc and polyclonal aLA were used. After washing, secondary antibodies were incubated for 1 h. For scFv-Fc, the anti-human Fc-HRP antibody was diluted 1:60,000, whereas anti-goat HRP was used for aLA in a dilution of 1:5000. The wells were washed, and TMB was incubated for exactly 10 min. H_2_SO_4_ was used to stop the reaction. The plates were then measured via spectrophotometry at A_450_–A_630_.

#### 3.2.9. Bead Conjugation

Bead conjugation was performed as described in [18]. Carboxy polystyrene beads had a size of 3.2 µm. To determine the surface loading of antibodies, the protein content of the supernatant after the conjugation was determined using the Pierce BCA Protein Assay Kit—Reducing Agent Compatible. Individual calibration curves were measured for each antibody, and the measurement was performed in 96-well plates. Therefore, measured antibody concentrations in the supernatant were subtracted from the total antibody amount initially added to determine the loading on the bead. Additionally, similarly to [18], beads were characterized via flow cytometry using secondary antibodies (anti-goat PE and anti-human Fc-PE), to confirm the conjugation.

#### 3.2.10. LPS Removal Assay in Water

Endotoxin solution (60,000 EU/mL) from *E. coli* O111:B4 was prepared in LAL water and left rotating for 15 min prior to the experiment. Beads were washed three times with LAL water by centrifugation at 5000× *g* for 5 min. Before use, the LPS solution was vortexed for 3 min. Subsequently, different Ab beads and unconjugated beads were resuspended with 60,000 EU/mL LPS and incubated for 20 min while rotating. After 20 min, samples were centrifuged at 3000× *g* for 5 min, and the supernatant was collected and further filtered in a centrifugal filter (0.65 µm). The endotoxin content in the flow-through was tested via the Pierce Chromogenic Endotoxin Quant Kit. The initial LPS solution, without beads, was properly diluted to fit the assay upper limit and was used as a positive control. More details are described in [18].

#### 3.2.11. Conjugated Ab Beads and Biotinylated LPS Measured in Flow Cytometry

Conjugated Ab beads and PS beads (10 µg/mL) were washed with endotoxin-free water and subsequently incubated with different biotinylated LPS concentrations (0; 1000; 5000; 10,000; 50,000; 100,000; 250,000; 500,000; 750,000; 1,000,000; and 2,000,000 EU/mL) in a volume of 100 µL for 1 h. The beads were transferred to FiltrEX 96-well filter plates and centrifuged at 3000× *g* for 5 min. Then, they were resuspended in 200 µL of PBS-F (10% FCS) and incubated for 10 min while shaking at 450 rpm. Subsequently, the beads were washed twice with 200 µL of PBS-F while always incubating the beads for 1 min and shaking at 450 rpm before centrifugation at 3000× *g* for 5 min. The beads were then stained with Streptavidin-APC (1:50) in PBS-F for 20 min in the dark at 4–8 °C. The beads were washed three times in PBS-F, resuspended in 100 µL, and transferred to a 96-well plate. The flow cytometry device MACSQuant X was used to measure the plate, and the MFI of each sample was analyzed using FlowLogic version 7.3.

## 4. Conclusions

In this work, we established a method of generating LPS-specific monoclonal scFv-Fc antibodies. Given that traditional antibody generation approaches like hybridoma technologies involve the immunization of animals, exposing them to a potential toxic shock when using LPS as an antigen, alternative methods excluding the use of animals were selected. Inspired by a previous work on phage display [28], we used a scFv phage library to screen for *E. coli* O111:B4 binders and expanded the method by implementing yeast display, a technique that was not yet used in connection with LPS. A major advantage of yeast display, when comparing these two techniques regarding the generation of LPS binders, is the absence of *E. coli*. To be precise, *E. coli* might interfere due to intrinsic LPS within the screening process in phage display.

Via phage display, two distinct binders (P1 and P2) were generated, one of which showed more specific binding properties during ELISA compared to a commercial polyclonal aLA antibody that binds to lipid A from *E. coli*. Using a biotinylated O-Antigen in our LPS targets, we introduce a more directed orientation for LPS molecules, favoring the selection of scFv specific for the core and lipid A regions. The yeast display generated 30 recurring binders of which we chose three (Y1, Y2, and Y3) for further analysis. Since *S. cerevisiae* lacks intrinsic LPS, a loss of possible binders due to interferences in the selection process with the host can be avoided. Furthermore, in contrast to immobilized LPS for the screening in phage display, binders were screened using LPS in solution, which increased the chance of finding differently oriented binders.

The five selected scFv-Fcs were coupled to PS beads and tested for their LPS-removal capacities. Conclusively, binders from yeast display removed higher amounts of LPS from aqueous solution (60,000 EU/mL) than binders from phage display. The LPS-removal assay showed that Y1-PS and Y2-PS outperformed aLA-PS, as the remaining LPS in solution revealed a difference by a factor of 265 and 53 times, respectively. This is in accordance with the trend observed in flow cytometry data, which generally showed higher quantities of LPS on the scFv-Fc-coated bead surface of yeast display antibodies.

Consecutively, it will be necessary to test the binding capacities of the selected binders not only in aqueous solutions of LPS but also in complex media, such as protein-containing solutions or pharmaceuticals. As shown using flow cytometry, yeast display-based binders were already able to capture large quantities of LPS. However, to establish a reliable assay, e.g., for LPS detection, the location of the binding site plays a crucial role in capturing LPS, regardless of its orientation. Instead of just one binder, a combination of different binders that cover various binding sites might ultimately lead to a universal assay.

## Figures and Tables

**Figure 1 ijms-25-02297-f001:**
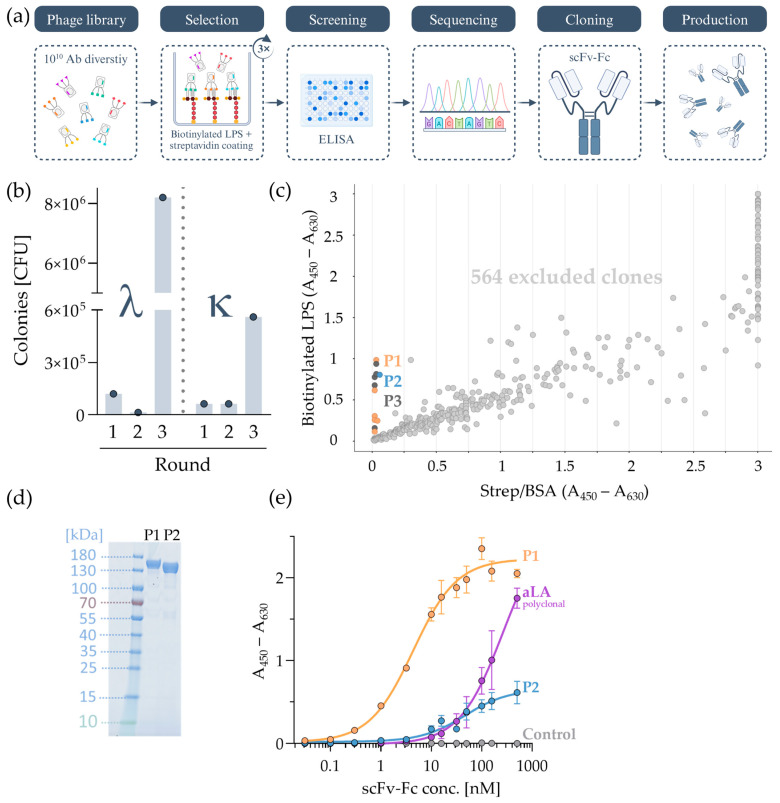
Generation of monoclonal antibodies targeting LPS from *E. coli* O111:B4 via phage display. (**a**) Schematic overview of phage display steps. In short, a scFv-phage library was used to select and screen for LPS-specific binders using streptavidin-coated plates and biotinylated LPS. Individual clones were chosen for their signal-to-noise ratio via ELISA. Followed by the identification of recurring binders achieved by sequencing, unique binders were then transformed into a scFv-Fc format and produced in Expi293F cells. (**b**) phage titer of panning rounds (1 to 3) for λ and κ sublibraries. (**c**) ELISA screening of 576 randomly picked individual clones. Clones that showed the lowest background signal (streptavidin/BSA) while displaying the highest LPS binding signal were chosen. Sequencing revealed recurrences. In orange: Clone P1, blue: Clone P2, and dark gray: Clone P3; dots in light gray represent excluded clones. (**d**) SDS PAGE gel of P1 and P2 showed bands related to scFv-Fc postproduction. Clone P3 was excluded since it showed streptavidin binding in higher antibody concentrations, as shown in Appendix A. (**e**) ELISA of scFv-Fc P1 and P2 in comparison to the polyclonal aLA antibody (in purple) and a non-specific scFv-Fc as a control (light gray). Data shown are mean ± SEM, *n* ≥ 3.

**Figure 2 ijms-25-02297-f002:**
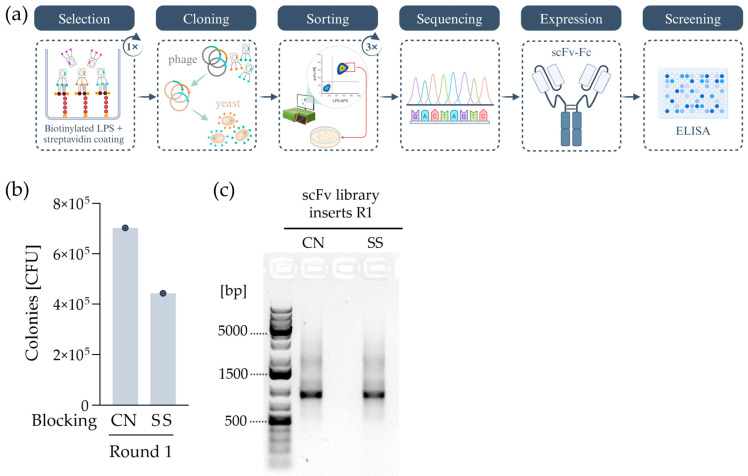
Generation of monoclonal antibodies targeting LPS from *E. coli* O111:B4 via a combination of phage and yeast display. (**a**) A schematic overview of selection steps. In short, after one round of phage display selection, plasmids of infected TG1 *E. coli* were isolated and enzymatically cleaved, and scFv genes were ligated into a *S. cerevisiae* display vector (pYSDM2). After transformation, scFv-expressing *S. cerevisiae* were sorted, aiming for the ones with the highest LPS-biotin binding. Then, single scFv clones were picked and sequenced. Recurring clones were transformed into a scFv-Fc format, produced in Expi293F cells, and screened via ELISA (Appendix A). (**b**) Phage titer was determined after the first round of phage display selection using a mixture of λ and κ libraries. Casein (CN) and sheep serum (SS) were used as blocking reagents. (**c**) Agarose gel of CN- and SS-blocked scFv library inserts.

**Figure 3 ijms-25-02297-f003:**
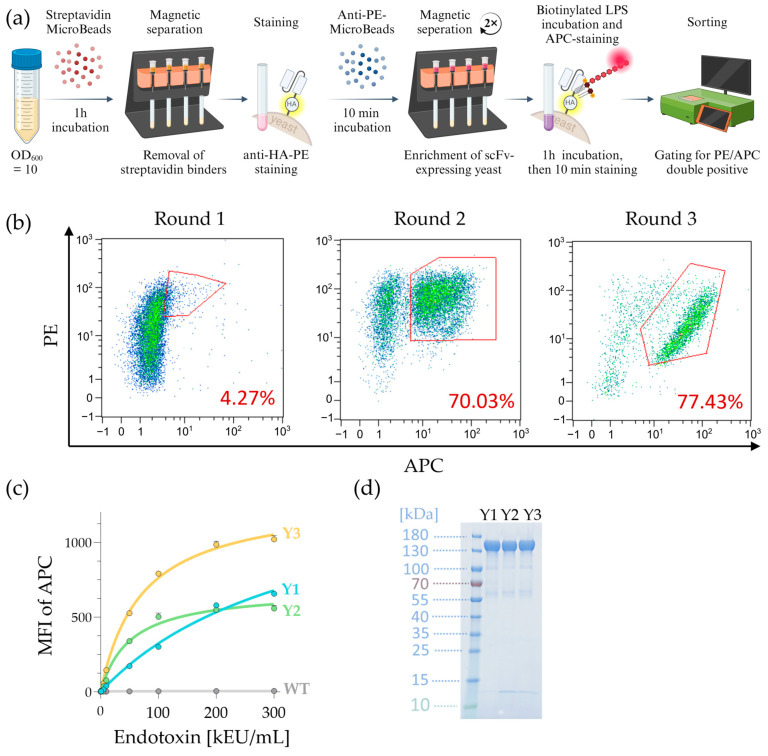
Cell sorting strategy to screen for LPS-specific scFv clones. (**a**) a schematic overview of *S. cerevisiae* library preparation before cells were sorted via MACSQuant Tyto Cell Sorter. First, yeast cells were incubated with Strepavidin MicroBeads for 1 h, and streptavidin-binding yeast cells were removed via magnetic columns. The remaining cells (flow-through) were stained with anti-HA-PE and subsequently incubated with anti-PE MicroBeads to magnetically capture all *S. cerevisiae* that express HA-tagged scFv on the surface. This isolation step was performed twice to enrich scFv-expressing cells. Enriched cells from the positive fraction were incubated with biotinylated LPS and then labeled with streptavidin-APC to sort for LPS-binding clones. Cells were transferred into a sterile Tyto HS (high speed) cartridge, and the gated population was sorted into the positive chamber. For rounds 2 and 3, the positive fraction was sorted an additional time. (**b**) cell sorting dot plots of initial stained scFv clones showed the frequency of LPS-binding yeast cells in the starting (round 1) and enriched fractions (rounds 2 and 3). Red gates indicated double-positive sorted cells that were enriched and used for culturing. To maximize LPS-binding clones for rounds 2 and 3, additional pre-selecting sorts (bulk sorts) were performed beforehand (see Appendix A). (**c**) the LPS-binding capacity of three yeast clones and their corresponding scFv that were chosen in the preselection (Appendix A) was determined by flow cytometry. Various concentrations of soluble biotinylated LPS were used to assess the MFI values of the resulting Streptavidin-APC staining. The titration curve represented clone Y1 in blue, clone Y2 in green, and clone Y3 in yellow. *S. cerevisiae* wild-type (WT) was used as a control and displayed in gray. (**d**) SDS-PAGE gel of purified Y1, Y2, and Y3 scFv-Fc.

**Figure 4 ijms-25-02297-f004:**
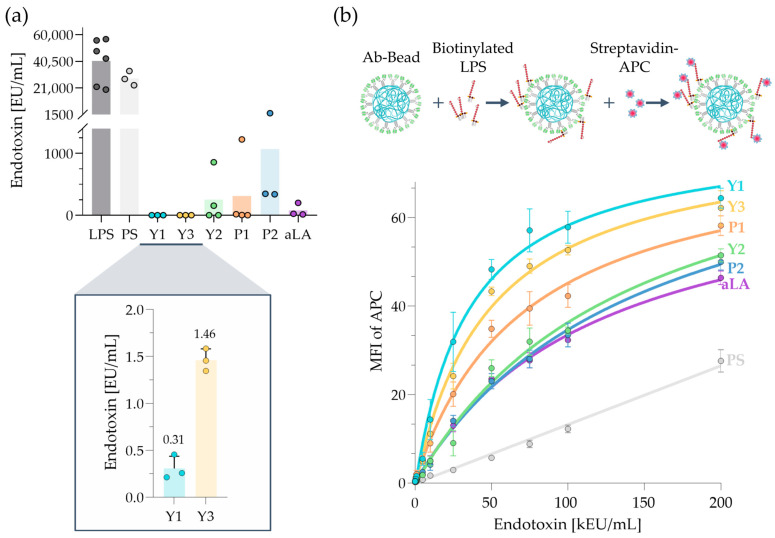
Polystyrene (PS) beads were functionalized with aminodextran and conjugated to the selected scFv-Fc. The resulting Ab beads were tested for their LPS-binding and removal capabilities. Polyclonal aLA Ab beads and unconjugated PS beads were used as positive and negative controls, respectively. (**a**) Beads were incubated with 60,000 EU/mL LPS from *E. coli* O111:B4 prepared in water to determine their LPS-removal capacity. The remaining endotoxin in the supernatant after the removal process was quantified via the traditional limulus amebocyte lysate test. Data shown are mean ± SEM, *n* ≥ 3. (**b**) Overall, 400-µg beads were incubated with different concentrations of biotinylated LPS from *E. coli* O111:B4 (0 to 200 kEU/mL) in a final volume of 100 µL. Streptavidin-APC was used to evaluate the binding of biotinylated LPS on the antibody-conjugated and control PS beads. Fitting curves (agonist vs. response (three parameters)) were calculated in GraphPad Prism version 8.2.1. The MFI was determined for the total population in three individual experiments.

**Table 1 ijms-25-02297-t001:** LPS removal (in %) after incubating 60,000 EU/mL LPS from *E. coli* O111:B4 with different Ab beads in water. Values were rounded to the second decimal place.

Ab Beads	Y1-PS	Y2-PS	Y3-PS	P1-PS	P2-PS	aLA-PS
LPS removal	100%	99.58%	100%	99.48%	98.22%	99.87%

**Table 2 ijms-25-02297-t002:** EC_50_ values of conjugated beads incubated with biotinylated LPS and measured through flow cytometry to assess the binding of the beads to LPS.

Ab Beads	Y1-PS	Y2-PS	Y3-PS	P1-PS	P2-PS	aLA-PS
EC_50_ (EU/mL)	34,958	141,571	48,538	72,319	150,713	117,854

**Table 3 ijms-25-02297-t003:** The PCR thermal cycler program that was used to amplify scFvs with homologous overhangs to the yeast vector.

Step	Temperature	Time	Cycles
Initial denaturation	94 °C	5 min	
Denaturation Annealing Elongation	94 °C 58 °C 72 °C	10 s 20 s 20 s	
			×35

Final extension	72 °C	5 min	
Cooling	12 °C	∞	

## Data Availability

The data presented in this study are available on request from the corresponding author.

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
