# Peer review of "Generation of Endotoxin-Specific Monoclonal Antibodies by Phage and Yeast Display for Capturing Endotoxin"

_ijms, 2024, doi:10.3390/ijms25042297_

Round 1

Reviewer 1 Report

Comments and Suggestions for Authors

Alexandra and colleagues have successfully generated antibodies targeted at endotoxins by utilizing a combination of phage display and yeast display techniques. The final single-chain variable fragment (scFv) candidates were effectively fused to an Fc fragment, resulting in scFv-Fc conjugates. These conjugates demonstrate significant binding affinity and selectivity, which could have widespread applications in endotoxin detection. I recommend that the paper be published, subject to minor revisions.

1.      It would be beneficial for the authors to include a brief introduction detailing the specific sequences incorporated in their library. This information would provide readers with a clearer understanding of the library's composition and its potential impact on the antibody's specificity and affinity.

2.      In relation to Figure 1d, it is unclear whether deglycosylation enzymes were used to remove glycosylation, as there appears to be no band smear under SDS-PAGE.

3.      The rationale behind opting for yeast display to screen the library following phage display could be more explicitly stated. Specifically, it would be insightful to understand if yeast display offers any particular advantages in the selection process for lipopolysaccharides (LPS), potentially contributing to the observed selectivity and binding affinity of the scFv-Fc conjugates.

Reviewer 2 Report

Comments and Suggestions for Authors

The manuscript sought to address the problem with methods of removal LPS by generating monoclonal antibodies specific against LPS. Fux et al. focused on generating monoclonal antibodies against E.coli LPS type O111:B4 by employing both phage and yeast display methods. Although the methods described in the studies were relatively new, the manuscript is very difficult to follow due to lack of details on methods used and background/ rationale on how the byproduct from the studies were superior.

Major comments:

1)    The flow cytometry data presented in the manuscript do not include how the major population was chosen. Forward versus side scatter (FSC vs SSC) gating is important to identify cells of interest based on size and granularity. Each dot plot here needs to include the FSC vs SSC to validate the gating and the comparison.

2)    In Figure 3 b, Rounds 3, the compensation is problematic.

3)    Table 1, how the purity of LPS is determined? Raw data is not available.

4)    The rationale of the study is not known. Why are yeast and phage display methods used? Why classic approach for generating monoclonal antibodies were not considered at all?  In discussion, it was not mentioned at all about how these methods are more efficient compared to others.

5)    Most of the figures were not generated based on results, although it might be necessary to include the schematic diagram considering the nature of the project.

6)    In Figure 3d, it is not clear how Y1, Y2, Y3 are different. If it is not, it is hard to understand why Fig 3c and d were shown.

7)    Page 6, Lines 210-212, it seems that the manuscript has not been revised or needs more work to do.

8)    Page 6, Figure 3d was mentioned before Figure 3c in Lines 232-235.

Comments on the Quality of English Language

Should be improved. 

Reviewer 3 Report

Comments and Suggestions for Authors
  • Major comments: 

The paper titled "Generation of Endotoxin-specific Monoclonal Antibodies by Phage and Yeast Display for Capturing Endotoxin" focuses on developing monoclonal antibodies to detect and capture lipopolysaccharides (LPS), a component of the cell wall of Gram-negative bacteria. This is significant because LPS can trigger severe immune responses and is a contaminant risk in pharmaceuticals and research products. The study compares the efficacy of yeast and phage display techniques in producing these antibodies. It finds that certain antibodies generated through these methods are highly effective in LPS removal, outperforming commercial polyclonal antibodies. This research has implications for improving LPS detection and removal, which is crucial for ensuring the safety of pharmaceutical products and laboratory research.

Overall, the paper contributes valuable knowledge to the field of biotechnology, particularly in the context of endotoxin detection and removal. While promising, the findings might require further validation in different settings or using various endotoxin types for broader applicability.

  • General concept comments

Here are some detailed considerations/suggestions for the study:

1.     While the authors noted the two scFv-M13 filamentous bacteriophage sublibraries, composed of different types of the VL domains (κ and λ) with an antibody diversity of 5 x 1010 was made in-house, more detailed information on the two scFv-M13 filamentous bacteriophage sublibraries should be disclosed, like how did you make the sublibraries, and why you chose these sublibraries for the screening over other libraries.

2.     While the authors measure the binding of P1, P2, Y1, Y2, and Y3 to LPS through Elisa and FACS, the exact values of the binding affinities (KD) should be measured of these Monoclonal Antibodies to LPS with proper controls.

  • Specific comments:

1)    Lines 68-69, the catalog number of polyclonal anti-lipid A (aLA) antibody should be noted in methods.

2)    Line 111-112, why do you generate monoclonal antibodies (scFv-Fc) against E. coli LPS type O111:B4, not another type of LPS?

3)    Line 134, how about the percentage of P1, P2, and P3 in the 12 clones?

4)    Lines 164-165, it would be better to disclose the EC50 values here.

5)    Line 211, Typo, Figure 3a should be noted here.

6)    Figure 3d, Multiple bands were observed for each line? Explanation?

7)    Line 266, please include this as supplementary data.

8)    Figure 4, from the result, PS also has some binding to Endotoxin (LPS). Besides, it seems like the MFI of Y1-3 or P1-2 didn't outbeat PS too much (only about 2-3 times higher).

9)    Line 472, Table 3 was not referenced in the main text. Besides, elongation was only 20 sec. How long is the PCR product?

10) Figure S4, not 288 total clones? Why do the signals of Y2 and Y3 decrease by higher concentration? Wired?

Comments on the Quality of English Language

Minor editing of English language required.

Reviewer 4 Report

Comments and Suggestions for Authors

Comments for the author of IJMS manuscript IJMS-2812205:

The author of the IJMS manuscript “Generation of Endotoxin-specific Monoclonal Antibodies by Phage and Yeast Display for capturing Endotoxin”, present their work toward solving the problems associated with detection and elimination of lipopolysaccharide (LPS).  Specifically, they now describe their method of producing LPS-specific monoclonal antibodies as single-chain variable fragments (scFv) using both phage- and yeast-display platforms.  These scFvs were then fused with human fragment crystallizable (Fc) for conjugation to microbeads.  The LPS removal capacity was then compared with that of a polyclonal anti-lipid A IgG bead.  Together, they selected 5 different LPS-specific scFvs that showed binding properties that were similar to the polyclonal antibody and two that outperformed the commercial product.  This shows that monoclonal antibodies against LPS can be designed using these two display platforms.  Below are some comments that I would like the authors to address as they revise the manuscript.   

General Comments:

  1. In the description of Figure 2 (line 181), casein is abbreviated as CS where it appears to be abbreviated as CN in the actual figure.
  2. Line 211 has an error message.

Round 2

Reviewer 3 Report

Comments and Suggestions for Authors

The revised manuscript "Generation of Endotoxin-specific Monoclonal Antibodies by Phage and Yeast Display for Capturing Endotoxin" effectively addresses most of my review comments. The authors have enriched the manuscript with additional background information, clarified methodological details, justified their choices, and improved language quality. They also addressed technical challenges related to binding affinity measurements. Given the comprehensive revisions and the manuscript's contribution to the field, I recommend accepting this paper for publication.

Comments on the Quality of English Language

Fine.